# Cost-Effective Seismic Exploration: 2D Reflection Imaging at the Kylylahti Massive Sulfide Deposit, Finland

**Suvi Heinonen [1],\* , Michal Malinowski [2] , Felix Hloušek [3], Gardar Gislason [1], Stefan Buske [3], Emilia Koivisto [4] and Marek Wojdyla [5]**

1 Geological Survey of Finland, Vuorimiehentie 5, 02151 Espoo, Finland; Gardar.Gislason@veitur.is
2 Institute of Geophysics, Polish Academy of Sciences, Ks. Janusza 64, 01-452 Warsaw, Poland; michalm@igf.edu.pl
3 Institute of Geophysics and Geoinformatics, TU Bergakademie Freiberg, Gustav-Zeuner-Str. 12, D-09599 Freiberg, Germany; felix.hlousek@geophysik.tu-freiberg.de (F.H.); stefan.buske@geophysik.tu-freiberg.de (S.B.)
4 Department of Geosciences and Geography, University of Helsinki, PL 68, 00014 Helsinki, Finland; emilia.koivisto@helsinki.fi
5 Geopartner Ltd., ul. Skośna 39B, 30-383 Kraków, Poland; marek.wojdyla@geopartner.pl
\* Correspondence: suvi.heinonen@gtk.fi; Tel.: +358-50-3488511

**Abstract:** We show that by using an advanced pre-stack depth imaging algorithm it is possible to retrieve meaningful and robust seismic images with sparse shot points, using only 3–4 source points per kilometer along a seismic profile. Our results encourage the use of 2D seismic reflection profiling as a reconnaissance tool for mineral exploration in areas with limited access for active seismic surveys. We used the seismic data acquired within the COGITO-MIN project comprising two approximately 6 km long seismic reflection profiles at the polymetallic Kylylahti massive sulfide mine site in eastern Finland. The 2D seismic data acquisition utilized both Vibroseis and dynamite sources with 20 m spacing and wireless receivers spaced every 10 m. For both source types, the recorded data show clear first breaks over all offsets and reflectors in the raw shot gathers. The Kylylahti area is characterized by folded and faulted, steeply dipping geological contacts and structures. We discuss post-stack and pre-stack data processing and compare time and depth imaging techniques in this geologically complex Precambrian hardrock area. The seismic reflection profiles show prominent reflectors at 4.5–8 km depth utilizing different migration routines. In the shallow subsurface, steep reflectors are imaged, and within and underneath the known Kylylahti ultramafic body reflectivity is prominent but discontinuous.

**Keywords:** seismic reflection; massive sulfide; seismic imaging; mineral exploration; cost-effective seismics; pre-stack depth migration; Finland; Outokumpu

## 1. Introduction

Seismic methods can be used to image subsurface geological features at high resolution down to several kilometers depth but are often considered expensive by the mineral exploration industry. The COGITO-MIN project (COst-effective Geophysical Imaging Techniques for supporting Ongoing MINeral exploration in Europe) investigated and developed cost-effective seismic exploration techniques [1] including passive seismic interferometry [2] and distributed-acoustic sensing vertical seismic profiling [3]. A major part of the cost of seismic surveys is attributed to the seismic source and the large field crews operating and moving the seismic receivers. With the advent of wireless receivers,

this technology is readily applicable to hardrock exploration, reducing the weight of equipment as well as the number of people required to operate it.

Here we discuss cost-effectiveness of active seismic reflection profiling for mineral exploration in crystalline rock environments. This includes the discussion about (1) the signature of Vibroseis vibroseis and dynamite sources, (2) the usage of different migration algorithms in data processing, and (3) the possibilities to record informative seismic data if source access is limited. The COGITO-MIN experiment took place in the Polvijärvi area in eastern Finland, where the polymetallic Kylylahti massive sulfide deposit is located. The survey site posed several challenges for the data acquisition, including inhabited areas, lakes, rivers, swamps, and dense forest. Vibroseis was the main source type used in the COGITO-MIN 2D survey but dynamite charges were also used in areas inaccessible for the Vibroseis trucks. Use of the two source types within the same area and survey motivates the comparison of Vibroseis and dynamite source signatures in hardrock mineral exploration.

Since the first hardrock seismic reflection surveys targeted mineral exploration in the 1990s (e.g., [4]), computing power has substantially increased and seismic data processing has been developed to better meet the challenges caused by typically complex geology and high seismic velocities in the hardrock environments. Kirchhoff prestack depth migration (KPreSDM) has proven to be an adequate imaging technique for seismic data acquired in crystalline environments, in particular for deep seismic reflection data [5]. Within Fresnel Volume Migration (FVM) [6,7], the standard KPreSDM approach has been further extended by introducing an additional weighting factor in the diffraction stack integral that focuses the back-propagated wavefield to its original diffraction/reflection point in the subsurface. FVM significantly reduces migration artifacts and as a result improves the final migrated image. Hlousek et al. [8] applied FVM to 2D seismic reflection data acquired in a crystalline environment at the German continental deep drilling site (KTB) near Windischeschenbach (South-East Germany). In the KTB drilling site, the FVM significantly improved the seismic images and revealed previously undetected structures in the seismic sections. In this paper, we compare the results achieved from standard post-stack and pre-stack time migration with the corresponding results from pre-stack depth migration (FVM), and also discuss the possible pitfalls in data processing that can cause loss of important reflections from the data.

The COGITO-MIN 2D seismic reflection data was acquired with dense source spacing because the importance of acquiring high-fold data has been emphasized in hardrock areas where reflection coefficients are typically low [9]. However, the sources are the major cost in a seismic survey and thus limiting their number could contribute to the cost-effectiveness of the method. Furthermore, while the deployment of receivers is often straightforward in terms of permits and accessibility, the situation is not the same for active seismic sources. Therefore, we also investigate how source decimation can influence the data quality and features imaged using the FVM approach.

## 2. Methodology

### 2.1. Geology and Survey Layout

The Kylylahti massive sulfide deposit is situated in eastern Finland within the North Karelia Schist Belt (Figure 1). Kylylahti belongs to the well-known Outokumpu mining district where mining and mineral exploration has been active since the discovery of the Outokumpu copper deposit in 1910 [10]. In 1908, geologists found an ore containing boulder some 50 km away from the bedrock hosted copper deposit that later developed into the Outokumpu mine. During the past hundred years, Outokumpu-type sulfide deposits have been discovered in a zone covering an area of about 4500 km$^2$. The Kylylahti polymetallic sulfide deposit was discovered in 1984 [11]. The deposit was intersected by drill holes guided by Mise-a-la-masse and EM frequency sounding results, as well as lithogeochemistry and geological models.

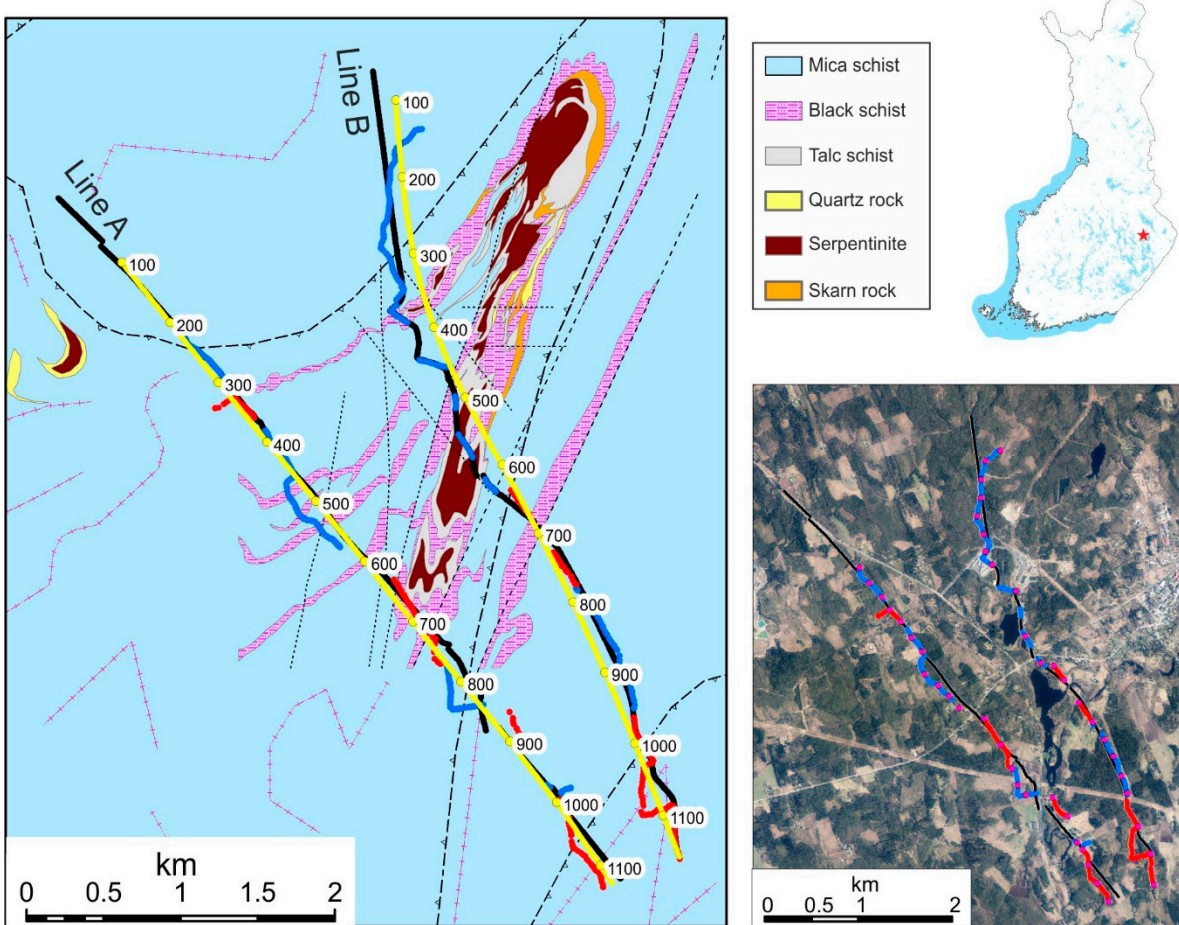

**Figure 1.** Layout of the COGITO-MIN profiles on a geological map [12] and on the aerial photo. The survey was done using Vibroseis (blue dots) and dynamite (red dots) sources at nominal 20 m spacing, and wireless receivers (black dots) deployed every 10 m. CDP points are plotted with yellow dots. Pink dots show the location of every 10th shot used in sparse shot stacking.

The bedrock of the Outokumpu region is dominantly mica schist that has some black schist intercalations. The massive sulfide deposits are associated with variably sized fragments and lenses of serpentinized mantle peridotites that have been tectonically emplaced upon and within the younger sediments [13]. The Kylylahti massive sulfide deposit is located at the margin of an ultramafic massif and mineralization is confined within the S–N trending eastern limb of the massif. In the Outokumpu district, Cu–Co–Zn–Ni–Ag–Au deposits are hosted by the so called Outokumpu assemblage rocks. This assemblage typically includes serpentinite, talc-carbonate rock, skarn, and quartz rock. This rock sequence is seismically reflective internally and especially against the mica schists motivating attempts to map the ore-hosting lithologies at depth using the seismic reflection method [14–16]. In the Outokumpu area, as well as in Kylylahti, the ultramafic massif is enveloped by carbonate–skarn–quartz rocks that in turn are surrounded by sulfide bearing black schist [17]. According to Rekola and Hattula [11], the Outokumpu assemblage rocks form a 3.3 km long and 400 m wide synform in the Kylylahti area. The synform dips at approximately 80° towards the west and the synform axis plunge is 33° towards south. The main mineralization consists of various separate lenses in the eastern limb of the synform.

The two COGITO-MIN seismic reflection profiles were acquired in an approximately NNW to SSE direction over the southern surface extension of the Kylylahti body (Lines A and B, Figure 1). The profiles are approximately perpendicular to the strike of the serpentinized Kylylahti rocks, and their location was mostly dictated by the gravel roads accessible to Vibroseis trucks. The COGITO-MIN

survey deployed a Wireless Seismic RT2 cable-less acquisition system provided by Geopartner Ltd. Each receiver station was equipped with a 10 Hz high sensitivity geophone, and planted at 10 m intervals along segmentally straight lines. The survey terrain varied from dense forest to fields and there were some gaps in the receiver line due to lakes and swamps. Telemetric data transmission between the receivers and recording truck enabled real-time data harvesting for QC purposes during the acquisition. Both seismic profiles had approximately 600 stationary receiver locations resulting in approximately 6 km long lines. Data acquisition parameters are listed in Table 1.

**Table 1.** The acquisition parameters of the COGITO-MIN 2D data.

| Data Acquisition | Line A | Line B |
|---|---|---|
| Receiver spacing (m) | 10 | |
| Source spacing (m) | 20 | |
| Vibroseis sweep (Hz) | 4–220 | |
| Number of sweeps/shot point | 3 | |
| Dynamite charge size (g) | 120 or 240 g | |
| Shot hole depth (m) | 1.5–2.5 | |
| Channels | 577 | 574 |
| Vibroseis source points | 121 | 152 |
| Dynamite source points | 98 | 85 |

We used both Vibroseis and explosive sources with a nominal 20 m shot interval. At each Vibroseis shot point, two INOVA UniVib trucks were used to produce three 16 s long linear upsweeps (4 to 220 Hz). These 9.5 ton Vibroseis trucks could be used on narrow forest roads but not along the full length of profiles. In order to avoid unnecessary gaps in the source coverage, we also utilized dynamite sources in the seismic data acquisition. Depending on the local condition, 120 or 240 g dynamite charges were placed in the predrilled holes of maximum 2.5 m depth. Because seismic survey lines bypassed some houses and other infrastructure, as well as lakes and swamps, some gaps in the source line were unavoidable.

*2.2. Fresnel Volume Migration*

The concept of Fresnel volume migration has been successfully applied to a variety of hardrock seismic data sets and it has significantly reduced migration artifacts and improved image quality [8,18,19]. Figure 2 illustrates the principle of Kirchhoff Prestack Depth Migration (KPreSDM) and Fresnel Volume Migration (FVM) with a simple synthetic model. It consists of a medium with a homogeneous background velocity of 5000 m/s and one diffraction point D located at x = 0.7 km and z = 0.3 km. The synthetic shot gather is calculated for a single source and 11 receivers. In KPreSDM, the recorded wavefield is distributed along the whole two-way traveltime isochrone ($t_0 = t_s + t_r$, where $t_s$ is the travel time from the source to diffraction point and $t_r$ is the travel time from diffraction point to the receiver) which results in a smeared image of the diffraction point for a single receiver. In FVM the emergent angle of the seismic wavefield is estimated at each receiver using a plane wave assumption. The semblance as a measure of wavefield coherency is evaluated at neighboring receivers for different potential emergent angles. The emergent angle with the maximum semblance value is used to trace a seismic ray from the receiver position back into the subsurface up to the two-way traveltime $t_0$. For the data presented here, 6 traces on the each side of the reference trace were used for local semblance estimate (total 13 traces). The smearing along the isochrone is restricted to the vicinity of the back propagated ray within its corresponding Fresnel volume, thus the image is focused to the relevant part of the two-way traveltime isochrone (Figure 2d). In KPreSDM the image results from constructive interference of the wavefield energy smeared along the isochrones for all receivers (Figure 2e). Although the smeared wavefield energy is concentrated at the diffraction point, strong migration artifacts around the diffraction point are still visible. These artifacts are suppressed in FVM and the image is better focused at the diffraction point (Figure 2f).

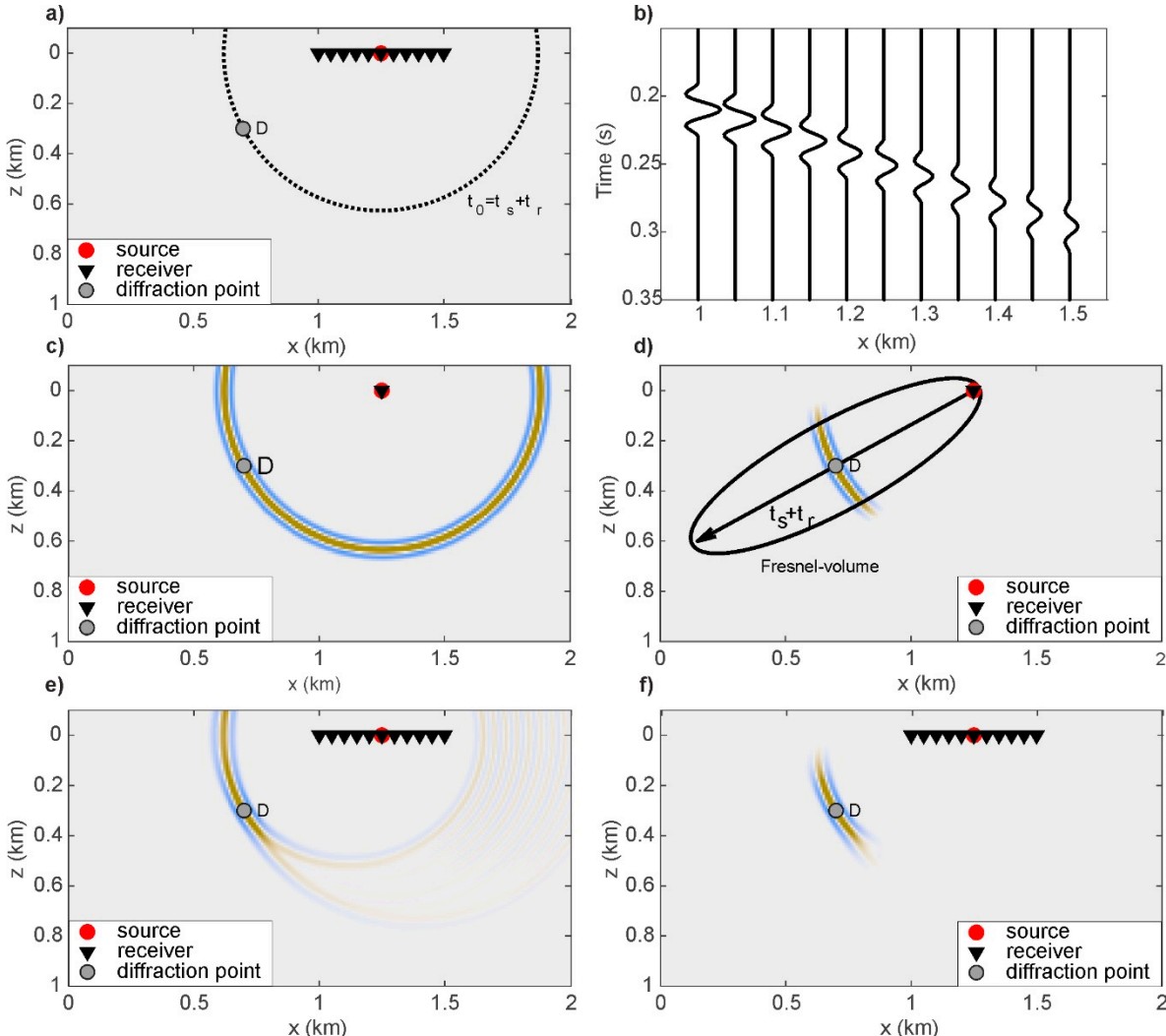

**Figure 2.** Principle of standard Kirchhoff prestack depth migration (KPreSDM) and Fresnel Volume Migration (FVM): (**a**) Model consisting of one diffraction point D embedded in a medium with a homogeneous velocity of 5000 m/s. (**b**) Synthetic shot gather for the source and 11 receivers in (**a**). (**c**) KPreSDM and (**d**) FVM images for a single receiver located at the source. The Fresnel volume is defined for a direct ray path starting from the receiver and propagated into the subsurface up to the two-way travel time of the corresponding time sample to be migrated (**e**) KPreSDM and (**f**) FVM images for all 11 receivers.

## 3. Results

Imaging of the COGITO-MIN data comprised the three main workflows, namely post-stack time migration (PoSTM), pre-stack time migration (PreSTM), and pre-stack depth migration based on Fresnel Volume Migration (FVM). In the following, these results are discussed.

### 3.1. Data Processing

The COGITO-MIN data pre-processing was common for all imaging methods used and included automatic gain control (AGC), bandpass filtering, deconvolution, and refraction static corrections. Additionally, the airwave and surface waves were attenuated with a median filter and first-arrival energy was muted using a top mute. The time processing flow is summarized in Table 2.

**Table 2.** Processing flow and parameters for the COGITO-MIN 2D data processing.

| Process | Parameter |
| --- | --- |
| Geometry setup | Crooked line; 5 m CDP spacing |
| Refraction statics | 2-layer model |
| Bandpass filter | 42-48-200-220 Hz |
| Airwave mute | 330 m/s attenuation |
| Median filter | Velocity-steered V = 3000 m/s |
| Deconvolution | Spiking 150 ms/Predictive 150/12 ms |
| Refraction mute | 50 ms below the first-breaks |
| AGC (Automatic Gain Control) | 250 ms |
| Velocity analysis and residual statics | 2 passes |
| NMO/DMO (Normal Moveout/Dip Moveout) | Integral DMO with 2nd pass velocities |
| Stack and F-K Stolt migration | V = 5400 m/s |
| Kirchhoff pre-stack time migration | With 2nd pass velocities |

Figure 3 shows Vibroseis and dynamite shot gather acquired at the same surface location before and after the processing described above. Both gathers are of good quality as demonstrated by the clear first breaks along the whole ~6 km long geophone spread and also by some reflections, e.g., at 600 ms. Figure 4 shows the frequency spectrum of the dynamite and Vibroseis shot gathers shown in Figure 3. As expected, the dynamite shot gather exhibits a higher frequency content, up to 420 Hz compared to the Vibroseis record. In the raw shot gathers, the reflectivity is mostly masked by the low-frequency surface waves which were effectively suppressed by band-pass filtering and a median filter. Reflections are imaged most clearly in the frequency band from 50 to 100 Hz, but the frequencies over 100 Hz also contain energy related to reflections as is seen in Figure 5. Surface wave noise is more prominent in the dynamite data while in the Vibroseis records noise is suppressed by the source array.

Deconvolution successfully decreased the reverberations of the first and later arrivals simultaneously increasing the high-frequency content of the data. A comparison of the Vibroseis and explosive sources show that the overall characteristics of the shot records are similar. Both records show the same reflections but the clarity of the reflections in the Vibroseis shot gather is enhanced due to the summing of three source sweeps which allows efficient suppression of noise. It is also notable, that the same channels are contaminated by noise in both shot records indicating identical data acquisition conditions. Furthermore, in both records noise in channels 40–45 is efficiently suppressed in data processing.

As usual in hard rock seismic exploration, static corrections play a crucial role in the visibility of reflectors [20–22]. In the Kylylahti region, the bedrock is overlain by glacial till of varying thickness that has a drastically lower velocity than the underlying bedrock. The variation of overburden properties and thickness causes time delays in the seismic signals originating from subsurface reflectors. The influence of such time delays can be clearly seen in the first breaks in Figure 3. Based on the automatically picked and manually quality controlled first arrivals, a 2-layer near-surface model was built. In the final models, the overburden thickness varied between 13 m and 56 m, average thickness being 36 m, and the average seismic P-wave velocity in the near surface layer was 2800 m/s. The corresponding static shifts ranged from –0.6 ms to 18.3 ms for receivers and –0.3 ms to 14.8 ms for shots, the average static shift being approximately 7 ms. The importance of the refraction static corrections are clearly demonstrated in Figure 6. If the refraction static corrections were not applied, the reflection at 1.5–2 km depth would not stack coherently.

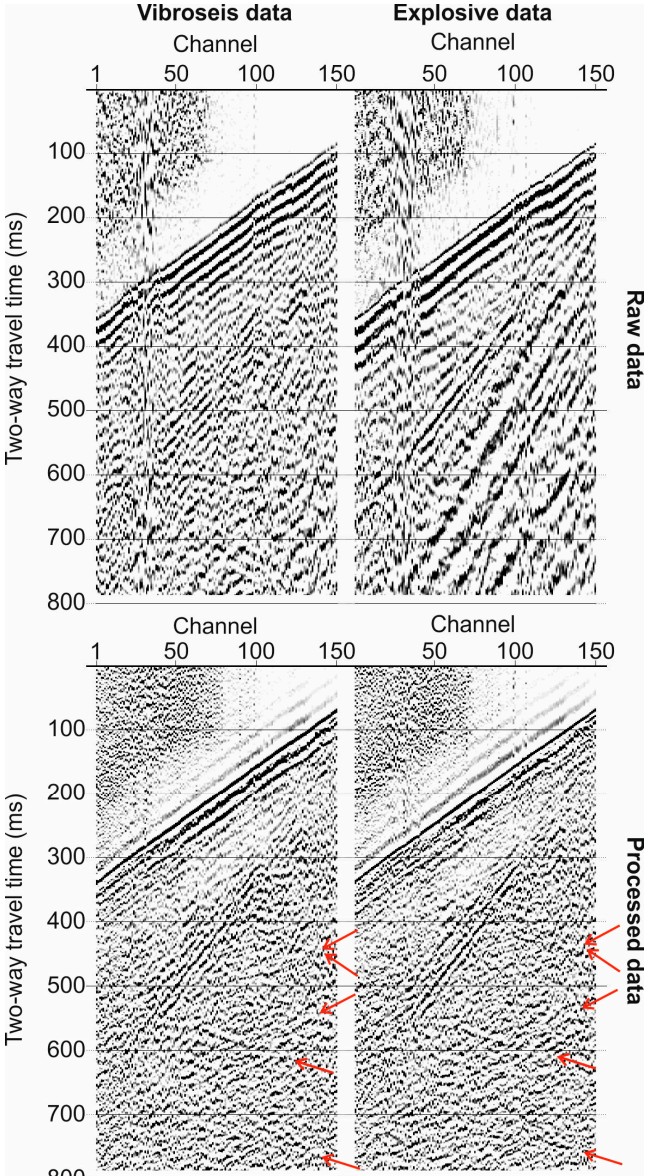

**Figure 3.** Vibroseis (**left**) and dynamite (**right**) shot gathers from the same location. Top row shows raw shot gathers with only automatic gain control (AGC) applied for plotting purposes, bottom row shows shot gathers after processing without top mutes applied. Several reflections (red arrows) emerge after noise suppression and static corrections, while deconvolution has successfully decreased the reverberations.

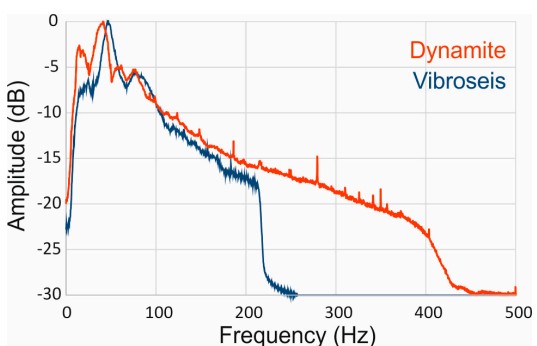

**Figure 4.** Frequency spectrum of the raw shot gathers in Figure 3.

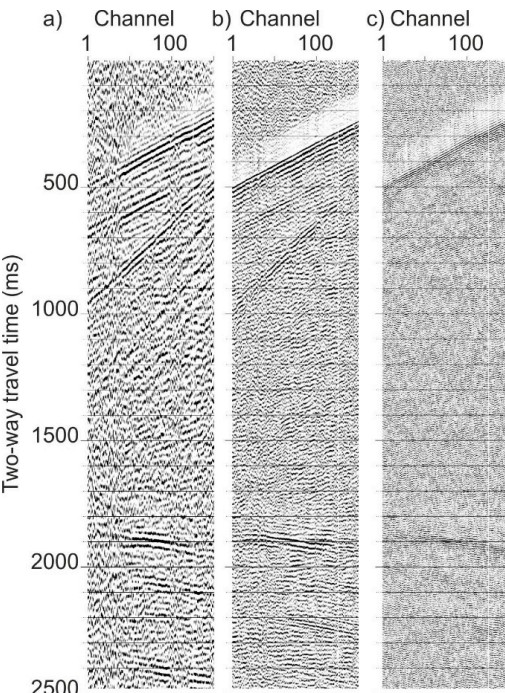

**Figure 5.** Bandpass filter panels for (**a**) 5-10-40-50, (**b**) 30-40-100-120 and (**c**) 90-100-200-250 Hz. Deep reflector (2000 ms) is visible in all frequency panels.

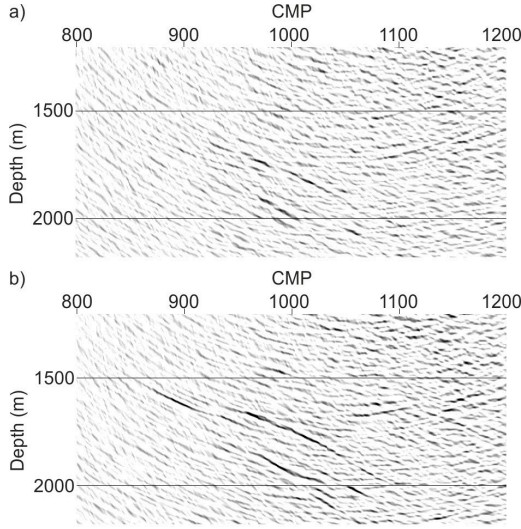

**Figure 6.** Example of a reflector in the post-stack time migrated seismic section with no refraction static corrections (**a**) and with static corrections applied (**b**). Influence of static corrections to imaging the reflector is significant.

### 3.2. Time Domain Imaging

After enhancement of signal-to-noise ratio of the shot gathers, we produced constant velocity stacks with post-stack time-domain Stolt migration in order to get an overall view of the subsurface reflectivity (Figure 7). The velocity field used for normal moveout (NMO) corrections was then refined but it was kept varying smoothly. We used a constant velocity of 5400 m/s for Stolt migration and for time-to-depth conversion. The velocity was chosen based on what had been used for previous seismic reflection data from the region [15]. The most prominent reflectors were imaged at 4.5–8 km depth (Figure 6). Previous publications have linked these strong reflectors to amphibolite sills on top of the Archean basement and they are also visible in all seismic reflection profiles previously acquired in

the Outokumpu district [15,23]. No coherent continuous reflections were observed at the location of the known Kylylahti deposit. The prominent but discontinuous reflectivity within the rock volume representing the Kylylahti ultramafic body is caused by tight folding of the ore-hosting rock types that have reflective near-vertical contacts with each other. In both reflection profiles, we observe prominent and continuous moderately dipping (approx. 20°) reflections at 500–1500 m depth southeast from the surface expression of the Kylylahti serpentinized ultramafic rocks.

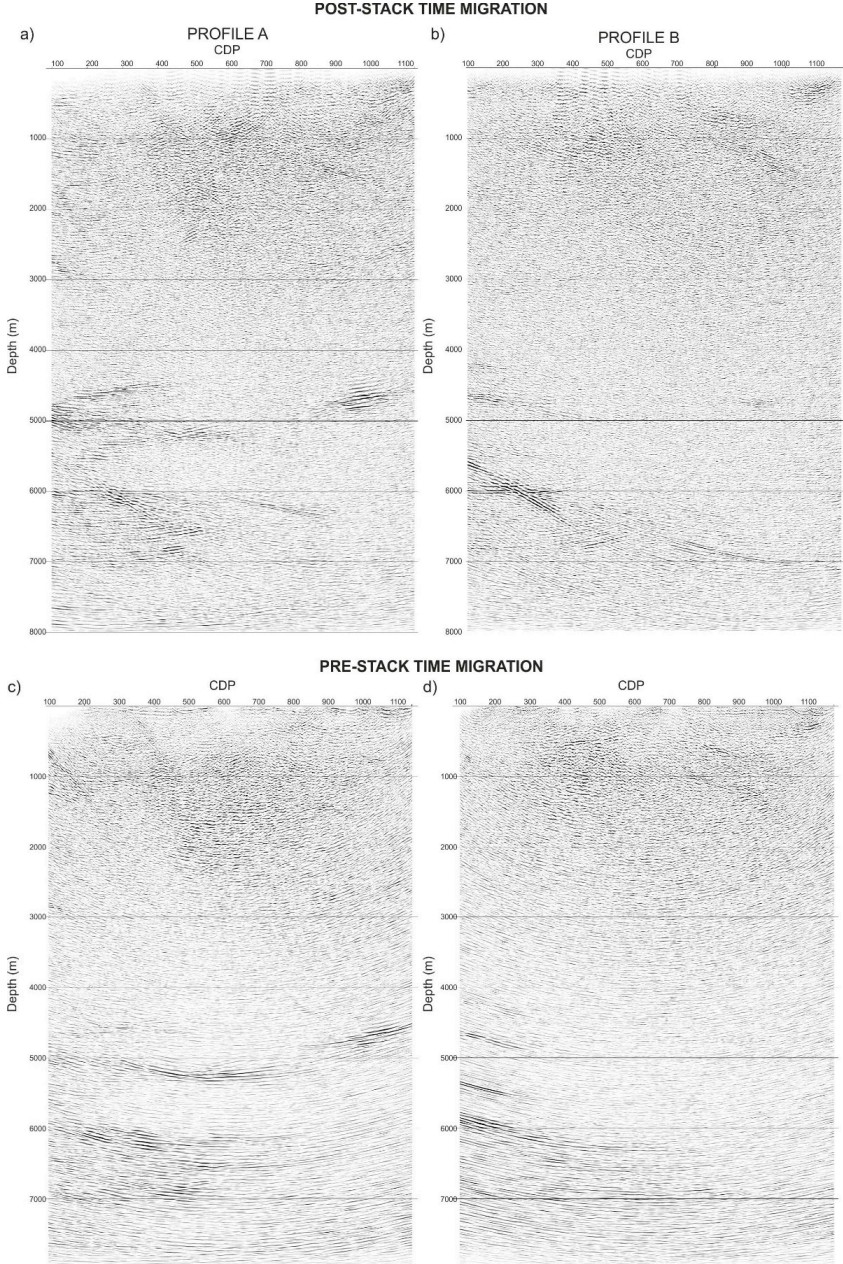

**Figure 7.** Comparison of the DMO-PoSTM results (**a**,**b**) with the PreSTM results (**c**,**d**) for lines A (left) and B (right). Strong reflectors are present at 4.5 to 7 km depth. In the shallow part of the seismic section, the reflectivity is discontinuous but prominent below the known surface exposure of the Kylylahti serpentinized rocks. Also more continuous, moderately dipping reflectors are imaged at 500 m to 1500 m depth between CMP 800–1000 on both lines A and B.

In the next steps, we attempted a Kirchhoff PreSTM using the same second pass velocities as in NMO-corrections of the PoSTM processing. We did not perform post-migration velocity analysis.

Imaging of the reflectivity down to ~1 km depth, which is of interest for exploration, was the most challenging issue. The source-generated noise masks shallow reflections independent of the source type, and additionally, near surface geological interfaces are subvertical and thus not ideal for surface seismic imaging. In the PreSTM results, a fairly steep reflector was observed reaching the surface at CMP 330. This is only observed in the NMO stacked data if unrealistically high stacking velocities are used for NMO corrections (Figure 8). Juhlin et al. [24] report a situation where dip moveout (DMO) corrections fail to produce a clear seismic image while use of high NMO velocity allows imaging of the steep dips. Also in the case of Kylylahti data, NMO velocity of 9000 m/s followed by the constant velocity Stolt migration enables the imaging of the same reflector as is seen in the PreSTM section. Later the efforts put on DMO parameterization paid off by making these steep reflectors visible in the final PoSTM sections without compromising imaging of the reflectors with more subtle dip and by using velocities corresponding to real rock properties. We used the time-domain Kirchhoff integral-based DMO on common-offset planes with the NMO velocities from the second pass of velocity analysis.

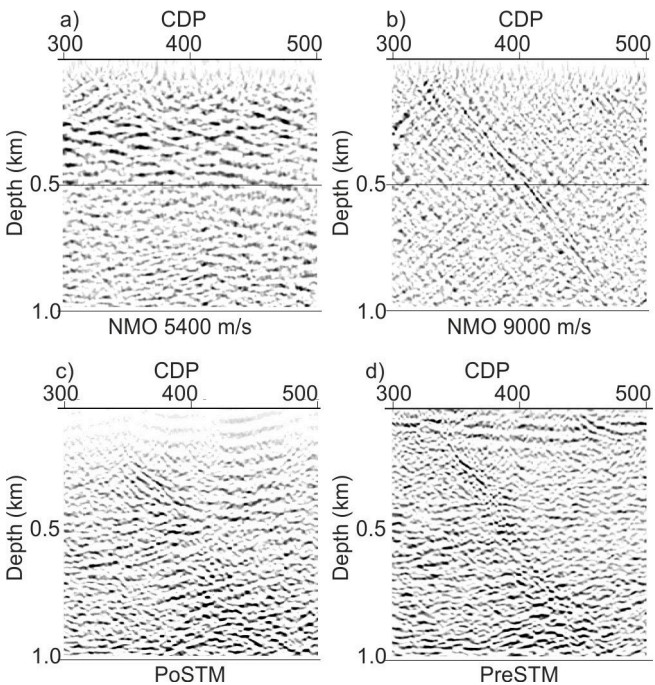

**Figure 8.** Detail of the seismic section A using (**a**) 5400 m/s and (**b**) 9000 m/s constant velocity for NMO correction followed by PoSTM and (**c**) DMO followed by the PoSTM. Figure (**d**) shows the PreSTM result of the same location. In the surface geological map, the reflector correlates with a black schist interlayer within the mica schist.

### 3.3. Depth-Domain Imaging with FVM

Since with the time-domain imaging we struggled to produce a meaningful picture of the target area down to 1–1.5 km depth, we decided to test the depth-imaging approach employing FVM. The data used for migration was pre-processed as described above. Unlike the conventional time-domain imaging approaches, which assume strictly 2D wave propagation and which are violated by crooked line seismic acquisition, the FVM algorithm used is implemented fully in 3D taking the true source and receiver positions into account. The migration grid was defined to extend 400 m longer than the actual receiver line on both ends also in order to capture the dipping reflectors from the edges of the seismic survey geometry that migrate out of the section. The grid cell size was defined to be 10 m in inline direction, 100 m in crossline direction, and 5 m at depth. A constant velocity of 5500 m/s was used for FVM performed on single shot gathers separately. The energy of the migrated single shot gathers was stacked to the final seismic image volume after FVM. By stacking the energy instead of the migrated

phases, the stacking result is less affected by velocity errors avoiding destructive interference. The FVM results show steep reflectors within the upper 1 km of the seismic section, strong but discontinuous reflectivity underneath the known Kylylahti body, and prominent reflectors at 4–7 km depth (Figure 9). Additionally, some dipping reflectors are imaged at the edges of the Kylylahti body. Figure 9 shows separate stacks for shot gathers acquired with Vibroseis and explosive sources. For both of the profiles, subsurface reflectivity characteristics are somewhat independent of the source type and distribution. The differences are discussed in detail in the following section.

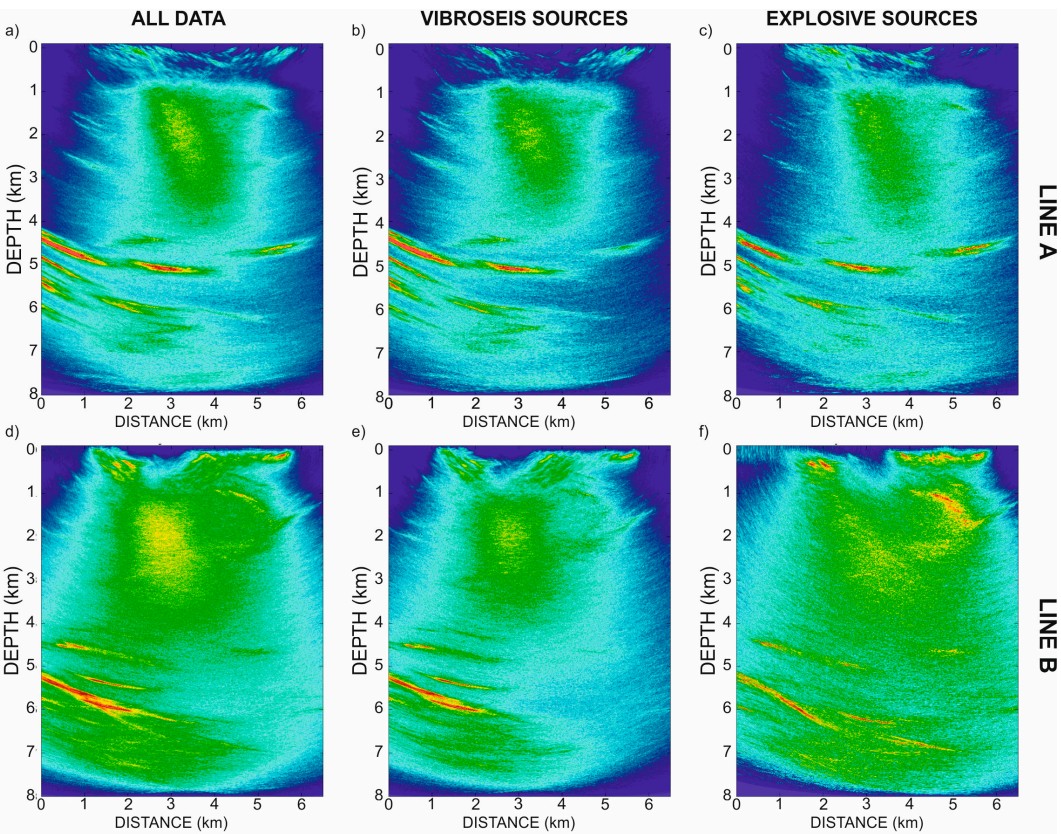

**Figure 9.** FVM results of the COGITO-MIN reflection seismic profile Line A (**a**–**c**) and Line B (**d**–**f**). The influence of Vibroseis (Line A: 121, Line B: 152 source points) and explosive sources (Line A: 98, Line B: 58 source points) to the final image of profiles A and B are separated in (**b**,**c**) as well as (**e**,**f**), respectively.

## 4. Discussion

The seismic source is the major cost factor in seismic reflection surveys and the use of active sources may also be problematic due to their environmental factors. For example, Vibroseis sweeps or explosions are not permitted close to infrastructure. Vibroseis trucks require a road or path to drive and permits are not always granted along public roads because of busy traffic. Additionally, mineral exploration takes place in remote areas, where no roads exist. Using Vibroseis sources in these kinds of conditions might be impossible, and the utilization of explosives that need to be drilled to a few meters depth cause economical, logistical or permitting issues. While these restrictions on source use are fixed, the seismic receivers are easily deployed in almost any kind of terrain, especially when modern wireless receivers are utilized such as in the COGITO-MIN seismic survey. Deploying receivers even in the middle of a city is often not a problem, but difficulty in achieving dense source distribution can sometimes prevent the seismic data acquisition. Passive seismic methods are currently being actively tested and developed to substitute active seismic sources but there is also a potential in using sparse seismic source distributions with relatively few active sources but a high number of receivers, in order

to achieve a higher resolution image than can be achieved with passive methods. We tested with the COGITO-MIN data and the FVM approach if a smaller number of source points could still reveal the relevant information about the subsurface reflectivity characteristics of the Kylylahti mining and exploration area. For this purpose, every 5th, 10th, and 20th shot gather was selected so that the source distribution was even along the profiles. The results are shown in Figure 10.

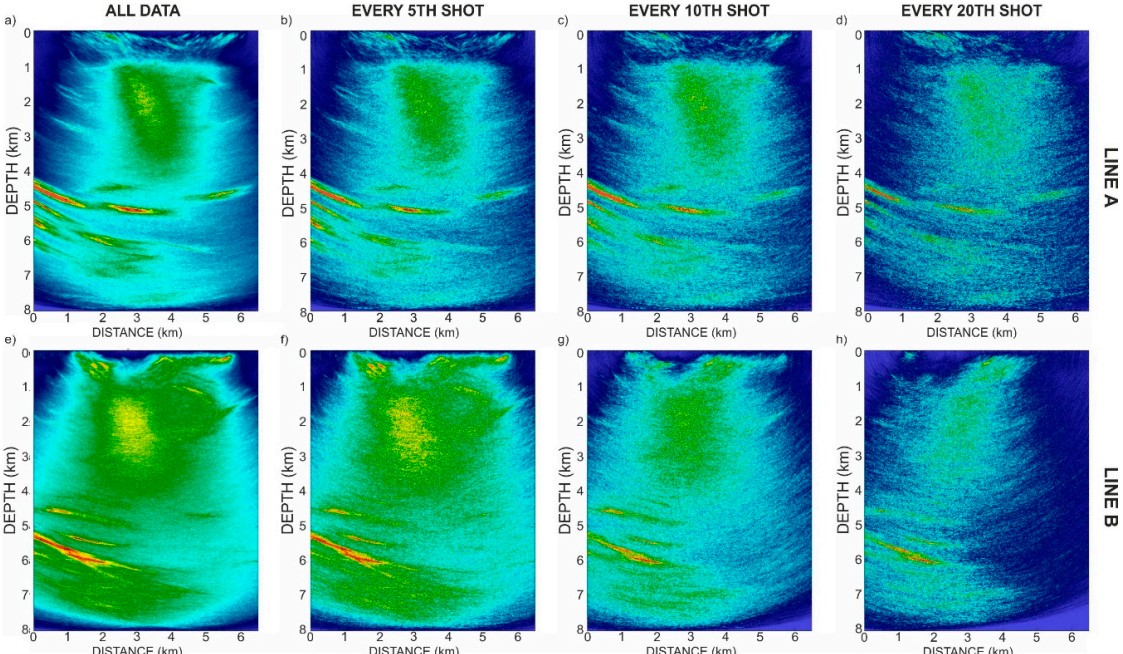

**Figure 10.** The FVM results of the COGITO-MIN seismic reflection profile A with (**a**) all (209), (**b**) 41, (**c**) 22, and (**d**) 12 stacked shot gathers, and profile B with (**e**) all (237), (**f**) 49, (**g**) 25 and (**h**) 13 stacked shot gathers. With decreasing number of shots the resolution of the image clearly decreases but still the main characteristics of reflectivity are visible in all images.

The PreSTM results with a limited number of shot gathers could not image reflections when every 5th source gather was stacked, and these results are not discussed further. However, the FVM results of the same data show that meaningful results could have been achieved with very sparse source spacing. In Figure 10, seismic data are stacked before FVM by using all, every 5th, every 10th and every 20th shot gather. The resolution of the images clearly decreases with the decreasing number of sources but only about 20 source points along the whole 6 km long profile appear to image a substantial part of the reflectivity observed in the full stacks. Logically, the near-surface reflectivity deteriorates because of lack of data while deep strong reflectors are imaged with only 12– 13 source points. The COGITO-MIN project also investigated the imaging power of sparse active seismic sources in the 3D layout as a side product of a passive seismic survey [25].

The results achieved with stacking only part of the shot gathers, migrated with the FVM approach, encourage the use of seismic reflection profiling also in areas where source accessibility is limited and only a few source points can be acquired. The COGITO-MIN sparse stacking test shows that even if the resolution of the images resulting from sparse sources is suboptimal, they still provide new and accurate information about subsurface distribution of the geological features. In our test cases, the nominal sparse source distribution was 120 m (every 5th) to 500 m (every 20th), while there was about 600 receivers on each line spaced every 10 m. However, we should note, that in those tests, the refraction static solution applied was obtained from the full dataset. We can expect deterioration in the quality of statics, when fewer sources are used.

The main cost factor in conducting a sparse source survey would be deployment of the receivers. The resolution of the achieved images would not be sufficient for exploration work aiming for target

generation but the sparse source seismic reflection survey combined with the corresponding advanced imaging approach could be utilized for example in green field exploration areas where the main characteristics of the subsurface reflectivity need to be confirmed or resolved prior to design of a full-scale 3D seismic survey.

In Figure 9, the contribution of the Vibroseis and explosive data to the FVM stack are shown. For line A, the explosive source points are more evenly distributed along the profile length than for line B (Figure 1) and thus also the resulting stacks from only Vibroseis and only explosive source records are more similar than in the case of line B, where most of the explosive sources are located in the SE end of the profile. In both cases, the main features are imaged with only Vibroseis or only explosive shot points although clearly the even and dense distribution of the sources results in the best resolution and clearest reflectivity. Based on these data, if there is no chance to conduct full scale seismic 2D profiles using Vibroseis sources, the explosive sources drilled to a few meters depth combined with efficient pre-stack depth migration algorithm such as FVM will provide informative insight to the deep subsurface.

Steep features imaged with the COGITO-MIN seismic reflection profiles were not revealed in the HIRE seismic sections previously acquired in the area [15]. This is likely due to the data processing that has not been fully optimized for the hardrock environment. In fact, our workflow was reversed: the steep reflectors were first imaged in FVM and only after this, did we have a closer look at the shot gathers and preserved them through the time processing/imaging flow by careful parameter selection. The FVM imaging was done using a constant velocity and with a fairly simple pre-processing sequence described earlier. Our results emphasize the robustness of the FVM method in producing an overview of the reflectivity when pre-hand knowledge of reflectivity characteristics or the detailed velocity field of the subsurface is poor. The steep reflectors might have been completely missed if only conventional time processing had been conducted for the data.

Steep reflectors in the shallow (<500 m) part of the COGITO-MIN seismic sections are imaged especially well in the FVM migration result (compare Figures 7 and 9) and are correlated with black schist interlayers in the mica schist environment in the geological map (Figure 1). Similar clearly defined, rather thin but more moderately dipping reflectors have been interpreted to originate from old fault zones that have been lubricated by the black schists [15]. The COGITO-MIN seismic reflection data provide an overview of the general geological structures in the Kylylahti area down to 8 km depth where prominent reflectors likely represent mafic sills above Archean basement. From the exploration point of view, the reflective features forming a thick package that could represent Outokumpu assemblage rocks are prospective while an individual reflector likely associated with a thin interlayer of black schist has no exploration interest. Unfortunately, imaging of the detailed folding and faulting pattern of the Kylylahti body is not possible by using surface seismic data acquired within the COGITO-MIN project because the dimensions of the geological features are too small compared to the seismic wavelengths used and the receiver spacing. However, the ore-hosting Outokumpu assemblage rocks manifest themselves in the seismic section by prominent but discontinuous reflectivity. Thus even if the individual layers and fold patterns cannot be imaged, the seismic reflection profiles shed light on the overall dimensions of the Kylylahti body.

## 5. Conclusions

The COGITO-MIN 2D seismic reflection survey comprises two ~6 km long profiles acquired with 10 m receiver spacing and using both Vibroseis and explosive sources with a nominal interval of 20 m. It appears that both source types produced almost equally good data although stacking of repeated sweeps within a source position improved the signal-to-noise ratio of the Vibroseis gathers. The equal quality of the sources is also demonstrated by results of stacking the two source types separately (followed by FVM); both resulting seismic sections image the most important subsurface features.

We tested three different imaging approaches; post-stack time migration, pre-stack time migration, and Fresnel Volume Pre-stack Depth Migration (FVM). The FVM was found to provide a robust

image of the main reflectivity characteristics of the subsurface, even with a constant velocity model. The advantage of such an approach is that it required only some basic pre-processing of the data. FVM results inspired us to revise our time-imaging workflow and only after that, were we able to produce comparable results with either pre-stack or post-stack time migrations.

The robustness of the FVM was also demonstrated with the decimated shots. We showed that by stacking only every 5th, 10th or even 20th shot gather we were able to retrieve the most prominent reflective features from the subsurface. These results encourage the use of the sparse source seismic reflection survey combined with efficient pre-stack depth migration FVM in mineral exploration areas where the use of active seismic sources is limited.

**Author Contributions:** Conceptualization, S.H., M.M. and E.K.; Methodology, S.H., M.M., F.H., G.G., M.W.; Software, F.H., S.B.; Validation, S.H., M.M., F.H., G.G.; Formal Analysis, S.H., G.G., M.M. and F.H.; Investigation, S.H., G.G., E.K., M.M. and M.W.; Resources, S.H., E.K., M.M., M.W. and S.B.; Data Curation, S.H., E.K., M.M. and S.B.; Writing—Original Draft Preparation, S.H., F.H.; Writing—Review & Editing, S.H., F.H., M.M., S.B. and E.K.; Visualization, S.H. and F.H.; Project Administration, E.K. and S.H.; Funding Acquisition, E.K., S.H. and M.M.

**Funding:** The COGITO-MIN project was funded within the ERA-MIN network. At the national level, the project was supported by Tekes (Business Finland) in Finland and National Center for Research and Development (NCBR) in Poland.

**Acknowledgments:** The COGITO-MIN project group and numerous people attending the project field work are cordially thanked for their contribution to the project success. The GLOBE Claritas software has been used for data processing. The Fresnel Volume Migration codes are developed in-house at TUBAF.

**Conflicts of Interest:** The authors declare no conflict of interest. The funders had no role in the design of the study; in the collection, analyses, or interpretation of data; in the writing of the manuscript, and in the decision to publish the results.

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
