# Peer review of "Cost-Effective Seismic Exploration: 2D Reflection Imaging at the Kylylahti Massive Sulfide Deposit, Finland"

_minerals, doi:10.3390/min9050263_

Round 1

Reviewer 1 Report

Dear Authors

I think this paper is well-written and after some moderate revision will be ready for publication. I am attaching a  marked pdf with my comments. There are some minor corrections that I marked in this file. As per main issues I summarize them here:

1. In Figure 1 please indicate in the map which line is A and which one is B.

2. In table 2, the low frequency cut at 42 Hz seems to be too high. Missing such a wide bandwidth of data could lead to multitude of imaging issues. Why did you use 42 Hz to high-pass the data?

3. in Figure 4, is Amplitude in db? If so please add that unit to vertical axis. How much usable are frequencies above 100 Hz? Could you plot an F-K plot of one shot gather to show if there is enough energy at higher that 100 Hz frequencies?

4. Line 241-245: This is a dangerous practice in my opinion. Considering the amount of coherent noise in your data, using such high velocities could easily pick coherent noises such as left-over surface waves, back-scattered noise, converted waves and show an anomaly! NMO by no means meant for dipping reflector imaging and even picking NMO velocities should be only focused on flat events. In fact, a single constant velocity stack section is never enough for NMO velocity analysis and you should use horizons, gathers, and semblance together to get a reliable NMO velocity. Then you can do best with DMO to fix the dipping layer effect. If you used unrealistic NMO velocity then you also ruined any chances of DMO working as well. No matter how much effort you put on DMO if your NMO velocities were unrealistic then the whole theory of applying NMO and DMO afterward is violated. I recommend removing such practice from your paper and also future projects.

5. Figure 7: I recommend removing this Figure and related analysis. Yes, in theory, dipping reflectors can make NMO velocity look higher but that velocity picking should be avoided and let DMO deal with dipping reflectors. That is why NMO needs to be picked using gathers and stacks and semblances that all indicate an enhancement of horizontal reflectors. NMO is designed for flat reflectors, not dipping ones. This practice of very high-velocity picking for NMO is dangerous in my opinion. Section (b) in this figure shows lots of crisscrossing parallel dipping events. The fact that in part one of them gets larger amplitude could be easily related to trace scaling issues in the data. Again, this practice of high-velocity NMO for detecting dipping reflectors is not reliable. The reference Juhlin et al. [24] have done something even more unrealistic. They have combined stacks from two different NMO velocities by a ratio of (40:60). It seems this kind of practice is more about being desperate to get dipping reflectors than real physics-based seismic data processing.

6. I think it would be nice to have a one-one comparison between FVM results and Post-stack and pre-stack migrated results for both lines. This will be 3 figure for each Line showing the image for the exact horizontal and vertical location.

Regards,

p { margin-bottom: 0.1in; line-height: 115%; }

Author Response

Thank you for constructive criticism and good suggestions for improving the paper. We have clarified the manuscript according to the suggestions made. Attached are our answers to the main issues.

Reviewer 2 Report

The paper is able to demonstrate dropping the number of sources in seismic acquisition can be compensated with advanced processing algorithms to achieve lower cost surveys. This however, was demonstrated using full dataset statics solution for partial source processing that can be considered a drawback for their analysis. Despite the statics factor in the processing, I still think the analysis is valid and valuable for publication in Minerals. 

I have added a number of correction pints to the attached pdf file that should be noticed by the authors before the publication. The main question is why low frequency portion of seismic measurements have been removed from the data in bandpass filtering (Table 2). I understand this was to suppress surface waves and shear waves that usually exist in that range but we have many tools to deal with them without loosing the low frequency energy.

with my regards

Author Response

Thank you for constructive comments. We have gone through the questions and suggestions in the manuscript and modified the text accordingly.

As reviewer points out, we have used band-pass filter to remove the surface and shear-wave related noise from the data. This is demonstrated in the filter panels in the new figure. It is true that the low frequency content of the data is lost, but we also argue that keeping the low frequencies does not improve the quality of our stacks –there is no additional reflectors imaged even if lower frequency content is kept and in the surface parts the wave lengths are too large compared to the scale of folding that they would produce meaningful image.

Reviewer 3 Report

This is a well written and well organised paper that demonstrates how seismic reflection methods can be used to image (partially) quite complex subsurface geological structures. Importantly the paper describes how the high costs of active source seismic surveys relative to other geophysical methods can be reduced. As such the paper can be of great value to the mineral exploration community. Given the emphasis on cost reduction, it would be helpful if the authors could provide an idea of the relative costs of these different approaches. I have no major technical criticisms, and attach an annotated manuscript with what I hope are legible suggestions for improvements to the written English. To demonstrate the application of the method, the authors might also highlight (label on the figures and speculatively interpret) some of the shallow features in the FVM images, e.g. X=1.8 km, Z=0.4 km and X=5 km, Z=1.2 km). I do have a few general comments that the authors may want to consider, but the manuscript could be published with quite minor revisions.

In Figure 2d, a Fresnel volume is shown for a scattered path that is stated as being from the source to scatterer and back to a receiver located at the source. I would expect the source and receiver to be at the ends of the Fresnel volume, which as drawn would imply the receiver is at 0.6 km depth. Maybe I have misunderstood something here, but would a half-volume be more representative for a reflection or scattered path? (For simplicity, the Fresnel volume and ts+tr ray path could just be removed).

Can you specify the source wavelet used for the modelling in Figure 2? Migration reduces the Fresnel volume to lambda/4, because the source and receiver are effectively downward continued to the scattering point.  But for a surface source and receiver all points within the Fresnel zone on a reflector contribute to the recorded reflection response, which I suppose is the argument for spreading the reflection out over a (surface) Fresnel volume after migration. This limiting of the smearing of the migration response is much of the value of the FVM method.

FVM migration has a couple of advantages, namely correct handling of the 3-D crooked line geometry and the migration of trace energy to minimise the effects of velocity error. Given Fig. 2 shows FVM of the data, what does the FVM result look like when trace amplitudes (+ve and –ve values) are input? Can we see a comparison with the results in Fig. 6? It is possible to carry out a comparable energy (+ve amplitude) migration with standard methods by first computing the trace amplitude envelope, bandpass filtering the data, and then running  a migration algorithm; this approach can produce significant improvement in the image quality, e.g. Nedimovic and West (Geophysics, 68, 286-296, 2003).

145-146: Over how many receivers was the local semblance estimated?

174: Stack is omitted from the processing flow in Table 2.

174/233: Be consistent in abbreviation for post-stack time migration (PSTM or PoSTM?)

239: Did you use a 2-D DMO, e.g. F-K, or 3-D DMO algorithm? If 3-D DMO, Kirchhoff DMO would smear that data along varying source-receiver azimuths and might improve the result with a crooked line geometry. Did you use a v(z) DMO algorithm?

243: What changes to the DMO parameters improved the result? Can you provide some detail?

Figure 6: I don’t understand why the PSTM and PreSTM results are so different with the PSTM appearing significantly less migrated than the PreSTM, especially when the migration velocities are meant to be the same. Note the crossing reflections at 6 km depth at CDP 250 in Fig. 6b. Can you explain these difference? Was the PresSTAM strictly 2-D with the same output image locations? Also, how was the conversion from time to depth carried out?

327: As the authors point out, the issue that isn’t really addressed by sparse acquisition is getting a good quality statics solution. So how would these FVM results look with a statics solution derived from a smaller number of shots? Perhaps too time-consuming to consider here, but a point for future study.

Author Response

Thank you for good comments and discussion.The answers to the questions are provided in the attaached file.
